# Revolutionizing Utility of Big Data Analytics in Personalized Cardiovascular Healthcare

**DOI:** 10.3390/bioengineering12050463

**Published:** 2025-04-27

**Authors:** Praneel Sharma, Pratyusha Sharma, Kamal Sharma, Vansh Varma, Vansh Patel, Jeel Sarvaiya, Jonsi Tavethia, Shubh Mehta, Anshul Bhadania, Ishan Patel, Komal Shah

**Affiliations:** 1Department of Information and Communication Technology, Dhirubhai Ambani Institute of Information and Communication Technology (DAIICT), Gandhinagar 382007, Gujarat, India; praneelsharmaxa@gmail.com; 2Department of Computer Science & Engineering, Ahmedabad University, Ahmedabad 380009, Gujarat, India; pratyushasharmaxa@gmail.com; 3Department of Cardiology, SAL Hospital, Ahmedabad 380054, Gujarat, India; 4GMERS Medical College and Hospital, Valsad 396001, Gujarat, India; vanshvarma2706@gmail.com; 5BJ Medical College, Civil Hospital, Ahmedabad 380016, Gujarat, India; vanshpatel433@gmail.com (V.P.); jeelsarvaiya26@gmail.com (J.S.); jonsi2002@gmail.com (J.T.); shubhmehta52@gmail.com (S.M.); anshulbhadania@gmail.com (A.B.); 6Department of Biology, Nova Southeastern University, Fort Lauderdale, FL 33328, USA; ip414@mynsu.nova.edu; 7Indian Institute of Public Health, Gandhinagar 382042, Gujarat, India; kshah@iiphg.org

**Keywords:** big data analytics, cardiovascular diseases, personalized medicine

## Abstract

The term “big data analytics (BDA)” defines the computational techniques to study complex datasets that are too large for common data processing software, encompassing techniques such as data mining (DM), machine learning (ML), and predictive analytics (PA) to find patterns, correlations, and insights in massive datasets. Cardiovascular diseases (CVDs) are attributed to a combination of various risk factors, including sedentary lifestyle, obesity, diabetes, dyslipidaemia, and hypertension. We searched PubMed and published research using the Google and Cochrane search engines to evaluate existing models of BDA that have been used for CVD prediction models. We critically analyse the pitfalls and advantages of various BDA models using artificial intelligence (AI), machine learning (ML), and artificial neural networks (ANN). BDA with the integration of wide-ranging data sources, such as genomic, proteomic, and lifestyle data, could help understand the complex biological mechanisms behind CVD, including risk stratification in risk-exposed individuals. Predictive modelling is proposed to help in the development of personalized medicines, particularly in pharmacogenomics; understanding genetic variation might help to guide drug selection and dosing, with the consequent improvement in patient outcomes. To summarize, incorporating BDA into cardiovascular research and treatment represents a paradigm shift in our approach to CVD prevention, diagnosis, and management. By leveraging the power of big data, researchers and clinicians can gain deeper insights into disease mechanisms, improve patient care, and ultimately reduce the burden of cardiovascular disease on individuals and healthcare systems.

## 1. Introduction

The term “big data analytics (BDA)” defines the computational techniques to study complex datasets that are too large for common data processing software, encompassing techniques such as data mining (DM), machine learning (ML), and predictive analytics (PA) to find patterns, correlations, and insights in massive datasets [1]. BDA has potential applications in healthcare to transform patient care and manage diseases by allowing personalized treatment plans and enhancing clinical outcomes [2]. Cardiovascular diseases (CVDs) are attributed to a combination of various risk factors, including sedentary lifestyle, obesity, diabetes, dyslipidaemia, and hypertension [3]. With mammoth datasets generated from electronic health records (EHR) and wearable health technologies, data sciences can identify trends and outcomes and personalize interventions. Apart from a better assessment of treatment efficacy, BDA not only reduces selection bias, but also, with the integration of wide-ranging data sources, such as genomic, proteomic, and lifestyle data, it could help understand the complex biological mechanisms behind CVD, including risk stratification in risk-exposed individuals [3]. Predictive modelling can help in the development of personalized medicines, particularly in pharmacogenomics; understanding genetic variation helps to guide drug selection and dosing, with the consequent improvement in patient outcomes [4].

The objective of this article is to evaluate and compare the available BDA algorithms along with AI, ML, and PL in the current era with their pitfalls and how these can be further integrated to refine CVD diagnostics and management.

To summarize, incorporating BDA into cardiovascular research and treatment represents a paradigm shift in our approach to CVD prevention, diagnosis, and management. By leveraging the power of big data, researchers and clinicians can gain deeper insights into disease mechanisms, improve patient care, and ultimately reduce the burden of cardiovascular disease on individuals and healthcare systems.

## 2. Big Data in Healthcare

The term “big data” refers to volumes of data that are too huge to handle using conventional software or web-based systems. Douglas Laney’s concept of three dimensions—volume, velocity, and variety, or the “*three Vs*” of (big) data is increasing [5] Variety refers to the various organized and disorganized data kinds that any company or system can gather, such as transaction-level data, video, audio, text, or log files. Velocity, on the other hand, describes the speed or rate of data collecting and making it accessible for further analysis. These three pillars now serve as the accepted definition of big data [6].

BDA is a complex process of reviewing large and varied amounts of data to identify hidden relationships, patterns, market trends, and other relevant information. This field uses different methods and tools, including data mining (DM), predictive modelling (PM), machine learning (ML), and artificial intelligence (AI), to process and analyse enormous amounts of data [7].

### 2.1. Enhancing CVD Treatment and Research Through Big Data Analytics

BDA uses large datasets, including genomic information, electronic health records (EHRs), and real-time health monitoring, to provide more comprehensive and detailed knowledge of cardiovascular illnesses. In healthcare, BDA can be helpful by using it as mentioned in Table 1.

Predictive algorithms have the ability to assess how each patient will react to a given treatment, which allows for the development of customized therapeutic strategies that maximize benefits and reduce side effects [8]. Furthermore, BDA can find new therapeutic targets and biomarkers by revealing patterns in huge datasets. Real-time modifications to treatment regimens are also made possible by the capacity to continuously monitor patient data using wearable technology and wireless mobile apps. This dynamic strategy maximizes therapy efficacy and improves patient outcomes by enabling prompt interventions and alterations based on ongoing data [9].

### 2.2. Challenges

BDA and management in healthcare have several obstacles. Health information is sensitive, making data security and privacy important issues. When combining various data sources, issues with compatibility occur, making it more difficult to synthesize accurate insights. Furthermore, the enormous amount of data necessitates significant processing and storage capacity, which is frequently expensive [10].

## 3. Applications of Big Data Analytics in Cardiovascular Diseases

BDA in the context of cardiovascular diseases (CVD) is a revolutionary tool for population health management, drug safety monitoring, personalized treatment, predictive modelling, and quality of care.

The combination of ML and AI has greatly improved predictive modelling. For example, retinal fundus photos have been proposed to identify cardiovascular risk factors, strengthening risk assessment and enabling early intervention options [11], exemplifying non-invasive imaging techniques with BDA to improve the predicted accuracy of cardiovascular risk assessment with models that analyse variables including demographic, clinical, and risk factors to stratify patients based on their risk profiles and reducing the incidence of cardiovascular events [12].

Various studies have proposed different models utilizing different inputs beyond BDA in prediction analysis for cardiovascular diseases, as shown in Table 2 [13].

Big data can be used to merge genetic information, lifestyle factors, and medical history to enable human–computer interaction and offer customized recommendations that enhance patient adherence and results [20]. This method is further improved by pharmacogenetics, which takes into account how genes affect drug responses [21].

Studies on the critical medications for CVD in low- and middle-income nations emphasize the significance of data-driven strategies for reducing healthcare inequities and gathering data from a variety of sources [22]. The ability to merge and evaluate the merged data of the mobile health technologies and biometric sensors is transforming population health management through the real-time tracking of cardiovascular health indicators, early intervention, and patient self-care including safety surveillance of elderly individuals’ prescription drugs [23]. The inclusion of empirical data in safety monitoring has been improving comprehension of drug efficacy across patient demographics. The study by Schaffer et al., 2022 [23] examines the hazards associated with polypharmacy use in patients treated with cardiovascular medicines and the optimization of their treatment plans and safety [24] (Jurgens et al., 2022). Big data help healthcare providers measure and improve the quality of cardiac care by analysing information about patient outcomes, allowing healthcare organizations to evaluate the effectiveness of treatments and identify areas for improvement. The potential of AI in personalized heart care has been emphasized for its role in improving care quality through data-driven insights [25].

## 4. Predictive Modelling and Risk Assessment

The technological advancements in the fields of ML, AI, and statistics, when implemented sequentially, can identify patients likely to experience cardiovascular events and help in suggesting appropriate therapies that can increase their survival rates.

An important application of ML is Artificial Neural Networks (ANNs) as depicted in Figure 1, being applied to symptoms and signs, drug dosing, the analysis of cardiac imaging, ECG signal interpretation, and the detection and management of coronary artery disease [26]. Simplistically put, based on the scientifically and statistically significant input datasets apart from the outliers, after being incorporated in the input layer, a weighted analysis identifies the contribution of each variable in the hidden layer, which, further in the weighted analysis in the output layer, gives output data as the presence or absence of cardiovascular disease, as shown in Figure 1. One such study described how effective machine learning is in predicting atrial fibrillation (AF) from EHRs and Cardiac Magnetic Resonance Imaging (CMRI), indicating how these models have helped to offer precise prediction to the patients on the basis of image and clinical findings [27]. The *QRISK4* and *PREVENT* models leverage EHR data combined with social determinants of health to predict 10- to 30-year cardiovascular risks. These models exclude race as a predictor and instead use social deprivation indices (income, education, housing) to address disparities [28]. A comparative cross-sectional study between ML and conventional mode of survival analysis showed that ML algorithms have a superior ability to predict cardiovascular risks, surpassing the conventional statistical approach. However, ML models may produce biased risk estimates if they fail to consider censoring in survival data [29].

An interesting study is the development of a system for predicting heart attacks that leverages user input to generate complete risk scores. This approach shows how AI-powered apps can leverage machine learning-based health predictors to manage cardiovascular health, using many risk factors and assessing them simultaneously. More importantly, big data from national administrative databases have been incorporated to predict cardiovascular risk through deep learning (DL) techniques, which reflect the importance of substantial data for improving the precision of risk evaluations [29].

ML algorithms can be used in EHR data as a technology to predict the occurrence of cardiovascular disorders [30]. Additionally, wearable technology like smart watches with built-in detectors and mobile applications enable the continuous monitoring of patient data, which further improves the potential for real-time risk assessment and can help in early intervention [31].

However, big data analysis has its limitations in its application to cardiovascular health. Despite excellent accuracy, ML can introduce extra complexity and obscurity in interpretation, making it very difficult for physicians to understand the factors behind the predictions [32]. Data-level approaches to address demographic biases and underrepresentation can be addressed with resampling by adjusting training data balance by oversampling minority groups or undersampling majority groups. 

Table 3 summarizes various models used for cardiac data analysis, detailing datasets, algorithms, features, and accuracy rates. Techniques like Learning Vector Quantization (LVQ), Recurrent Neural Networks (RNN), Multilayer Perceptron (MLP), and Classification and Regression Tree (CART) are applied to datasets such as UCI, Cleveland, and Framingham. Accuracy rates range from 71% to 98%, highlighting the efficiency of different approaches like LVQ and Multinomial Logistic Regression (MLR).

## 5. Personalized Medicine

Personalized medicine, or precision medicine, is a revolutionary change in healthcare that concentrates on customizing medical treatments according to the unique characteristics of each patient. This method differs from the traditional “one-size-fits-all” approach. In the field of personalized medicine, the utilization of BDA has created opportunities for innovative and targeted treatment approaches. These tactics rely on combining genomics, precision medicine, and advanced data analytics to enhance the precision of diagnoses and outcomes. The Apple Heart Study analysed data from 400,000 participants using smartwatches to detect irregular pulses. This wearable-derived data, when integrated with EHRs for validating atrial fibrillation detection, achieved 84% concordance with clinical ECG findings [37].

### 5.1. Role of Genomics and Precision Medicine

Personalized medicine has paved the way for FDA-approved treatments, targeting the specific characteristics of individuals, including not only a person’s genetic makeup but also the genetic profile of their tumour. Patients with many cancers now have access to molecular testing, allowing physicians to select treatments that improve survival chances while decreasing exposure to adverse effects [38]. However, the greater integration of faster processors, larger memory, and highly advanced algorithms, methodologies, and cloud computing may become critical for the continued existence of clinical information [39].

Based on the genetic information from a patient, it can establish genetic propensities for specific cardiovascular diseases, such as coronary artery disease or atrial fibrillation. For example, certain gene variants result in polymorphism in how individuals metabolize selected medications. This gives rise to pharmacogenomics, in which medications are genetically tailored for the patient, aiming for efficacy maximization while minimizing adverse effects. Genomic epidemiology using an integrated approach in groups representing categories for cardiovascular conditions with many genes and single nucleotide polymorphisms (SNPs) has identified elevated cholesterol and other lipid disorders, hypertension, and blood pressure in atherosclerosis and myocardial infarction [40]. Any given SNP is found on virtually every 1000 base pairs with approximately 3 billion base pairs of the human genome sequence, and there are likely >10,000,000 or an even greater number of SNPs that may make up the variation to enable SNP genotyping [41].

Network analysis carried out by utilizing a precision medicine perspective is capable of clustering individuals purely on endophenotype. It optimizes medication and behavioural changes that improve health, disease prediction and prognosis, disease biomarker identification, clinical trial enrolment enrichment, and the development of exposure that can optimally be tailored to the patient [42].

The BDA approach can break through the limits of traditional genome-wide association studies (GWAS) in engaging the propensity of understanding genetic variances and their interactions with environmental factors. For large-scale GWAS, there has been a large number of genetic loci significantly related to blood pressure and hypertension, and the Exome Aggregation Consortium have found the phenomenon in which some mutations such as those in the BMPR2 gene are associated with pulmonary arterial hypertension with variable penetrance. By integrating multiple data sources and employing novel analytical frameworks, big data analysis progresses towards precision medicine with targeted approaches and treatment modalities for cardiovascular diseases [43].

### 5.2. Tailoring Treatments Based on Individual Data

Various treatment modalities for cardiovascular disease are being aimed at as personalized medicine via Nanobiotechnology and Stem-Cell Therapy.

Nanoparticles are known to possess dual functions, offering both molecular imaging and therapeutic delivery. Most commonly, magnetic nanoparticles (MNP) such as contrast agents in MRI are said to detect high-risk plaque in atherosclerosis and show active vascular inflammation or thrombosis [44]. Such nanoparticles may be modified with therapeutic agents such as plasmid DNA for gene delivery or to allow targeted therapy verified through imaging modalities [45]. Additionally, perfluorocarbon nanoparticles could unite molecular imaging with localized drug delivery or infuse an enhanced targeting approach to therapeutic interventions. The development of lncRNA AK083884 and nanozyme-enhanced tyramine signal amplification probes have shown protection from viral myocarditis, and its genetic implications can have futuristic value in preventing the same [46,47].

Conversely, personalized cell therapy offers the possibility to obtain iPSCs from a patient’s somatic cells, allowing this personalized approach to create cardiomyocytes and bioengineered pacemakers adapted to individual disease profiles. These advances show how individualized data transform therapeutic strategies into precision treatment options in cardiovascular care [48].

AI techniques include LoRA, quantization, and several others. The former is a regression by which complicated machine learning models are made simpler. For example, these AI models, which make predictions regarding how patients respond to treatments for CVDs, can further develop LoRA’s optimization capabilities to effectively mitigate the difficulty in analysing extensive datasets that may enhance clinical decision-making. Together, LoRA and quantization make probable individualized treatments that are not only precise but widely available, improving the management of CVDs across settings [48].

Familial hypercholesterolaemia is a monogenic disease characterized by slightly elevated levels of LDL-C, with the early development of atherosclerosis and consequent cardiovascular disease in youth. The mutations in the LDLR gene can be of two types: homozygous (HoFH), where there are functional deleterious mutations in both alleles, or heterozygous (HeFH), involving deficiencies in one allele. Inhibitors of PCSK9 in patients with residual LDLR activity are represented by monoclonal antibodies—Evolocumab and Alirocumab—which reduce LDL-C levels through the inhibition of PCSK9. In HoFH patients bearing null mutations in a single LDLR allele, ANGPTL3 inhibitors such as Evinacumab reduce LDL-C substantially. Inclisiran, as a small interfering RNA (siRNA), allows the administration of a less frequent dosing schedule by inhibiting hepatic PCSK9. Their capability to individualize treatment based on genetic profiles and improve the treatment outcomes for patients with FH represents another landmark towards personalized medicine [49].

### 5.3. Using Big Data for Public Health Initiatives

The goal of ML and AI is to develop models that learn from multiple data types to make predictions, similar to human cognitive functions. Multimodal ML combines multiple data types like image, text, and speech to create more robust models. The process of learning representations based on multimodal input sources is called multimodal learning [50]. The process of merging information from several measuring modalities—such as imaging, text, and genetic data—in the context of cardiovascular illness is known as data fusion. Recent evidence highlights the application of multimodal learning and data fusion in improving risk stratification and decision-making for cardiovascular diseases. Researchers combine genomic data, cardiac imaging, and EHRs to identify high-risk individuals for major adverse cardiovascular events (MACE) with unprecedented accuracy. For instance, the UK Biobank34 and the Million Veterans Program 35 provide integrated genetic markers and quantified imaging studies with Electronic Health Records (EHRs) that may be utilized to enhance prognostic and therapy response prediction for cardiovascular illness [51]. Moreover, AI-driven systems trained on large-scale datasets like the UK Biobank can offer personalized recommendations for therapeutic interventions by predicting the likelihood of specific responses to treatments.

### 5.4. Identifying Trends and Patterns in CVD

Risk Prediction Models: Traditional cardiovascular risk prediction models use only a few clinical parameters such as age, sex, blood pressure, and cholesterol levels. BDA can make the models more powerful by incorporating increasingly large pools of data including imaging and electronic health records (EMRs). Chaves et al. developed an advanced risk prediction framework by combining deep learning techniques with abdominopelvic CT imaging features and EMR data. Their approach outperformed traditional risk scores for the prediction of ischemic heart disease (IHD). Higher predictive power could be achieved because the model integration of imaging and EMR could reveal interactions and correlations that were not obvious before [52].

### 5.5. Improving Disease Detection and Diagnosis

**a.** **Acute Cardiovascular Disease Detection**: Pattern recognition via BDA makes it possible to detect acute cardiovascular diseases in early stages with high accuracy. Zhang et al., [53] proposed a multimodal-based strategy by fusing ECG, phonocardiograms, echocardiography, Holter monitors, and biological markers for CAD detection, reaching high diagnostic accuracy based on the complementary information among different data modalities [52].**b.** **Severity Assessment**: The appraisal of cardiovascular disease severity can be greatly improved via the integration of several imaging techniques [52] by combining echocardiography and cardiac MRI to boost the prediction of sudden cardiac death in dilated cardiomyopathy patients. This multimodal method reaches a more complete evaluation of cardiac function and structure and increases accuracy in severity appraisal [54].**c.** **Early Identification of At-Risk Populations:** BDA also enables the early identification of populations at risk for CVD by utilizing lifestyle, genetic, and environmental data. Studies have demonstrated the integration of wearable device data, genomics, and social determinants of health to identify high-risk individuals for atrial fibrillation. This proactive approach facilitates timely preventive interventions, significantly reducing the burden of disease and healthcare costs

In the study by Chaves et al., a segmentation model alone, an imaging model alone, a clinical model alone, and fusion model-based machine learning models were developed and validated in a 5-year ischaemic heart disease prediction task. The “Imaging + Clinical Fusion” model identified more people at high risk of future 5-year ischaemic heart disease when compared against standard of care tools such as Framingham Risk Score (AUC 0.862 vs 0.776) [55]. Illustrating the potential of multimodal data fusion for reducing cardiovascular disease risk as well as early detection, Ali et al. used data from mobile and medical sensors (i.e., blood pressure, ECG, EMG), electronic medical records (EMR), and other relevant sources for multimodal fusion to develop an ML model capable of automatically giving recommendations for cardiovascular treatment. The important phases are outlined in data collection, feature extraction, feature fusion, and prediction processing by applying deep learning techniques. Before using deep learning mechanisms for disease detection and prescription plans, firstly, sensor and EMR data were fused via a feature matrix. Several preprocessing steps such as normalization and feature selection have demonstrated how this can help make the cardiovascular disease care process more accessible and precise by generating personalized recommendations based on patient data acquired from wearable low-cost sensors or other devices. They also show the disruptive changes made by big-data analytics and machine learning in advancing patient disease management and prescription plans [56].

## 6. Drug and Medical Device Safety Surveillance

One essential component of post-market surveillance is tracking negative effects. The Manufacturer and User Facility Device Experience (MAUDE) database maintained by the FDA is the main source of adverse event reports; however, because it relies on passive reporting, there may be underreporting and a dearth of thorough exposure data [57]. Research suggests that the incorporation of unique device identifiers (UDIs) into electronic health records (EHRs) may improve the monitoring of adverse events related to devices, which in turn may enhance the identification of safety signals [58].

Furthermore, active surveillance methods, such as the Data Extraction and Longitudinal Trend Analysis (DELTA) network, have demonstrated promise in detecting new safety issues via automated signal detection [59]. Real-time data analysis has emerged as one of the most important tools in medical device safety surveillance in recent years. The FDA’s Sentinel Initiative is one example of this approach, which uses electronic health data to monitor drug and device safety [60]. Emerging technologies, such as natural language processing (NLP) and machine learning algorithms, are now being used to analyse unstructured data within EHRs, such as physician notes and patient narratives, offering enhanced capabilities for detecting safety signals beyond traditional systems like MAUDE [59].

Case studies show how real-time data can inform safety assessments and enhance patient outcomes, such as those involving endovascular aneurysm repair devices, highlighting the usefulness of linked registry claims data in long-term surveillance [61]. For example, registry-based active surveillance, as demonstrated by Resnic et al. [62], successfully linked patient registries with claims data to detect safety issues with endovascular aneurysm repair devices, directly influencing clinical guidelines and improving outcomes. Numerous studies provide instances of effective surveillance methods. For example, a survey of medical device manufacturers found that single-arm trials and registries are used for most of the post-market clinical research and that these methods frequently lack the statistical power required for thorough safety evaluations [63].

Furthermore, the examination of incidents involving ventilators has demonstrated that improving healthcare personnel’s knowledge and training can greatly lower adverse events [64]. In summary, a multimodal strategy that incorporates case studies to guide best practices, real-time data analysis, and the strict monitoring of side effects is needed for efficient drug and medical device safety surveillance.

## 7. Quality of Care and Performance Measurement

In cardiovascular medicine, using comprehensive data analytics aids the ongoing evaluation of quality through various performance metrics. Common indicators include patient death rates, adherence to clinical guidelines, hospital readmission frequencies, and the careful use of treatments like percutaneous coronary intervention (PCI) within specific timeframes. Newer indicators have appeared such as tracking frailty in heart failure patients incorporating EHR data and using machine learning to spot patterns linked to poor outcomes. A systematic review showed that hospitals that use data analytics platforms to monitor real-time adherence to clinical protocols saw better results in both death and readmission rates, showing how data-driven monitoring improves overall care quality [64].

A key study looked at using neural networks to predict the risk of gestational diabetes mellitus (GDM). The researchers examined a dataset from the National Institute of Diabetes and Digestive and Kidney Diseases (NIDDK) using a radial basis function network (RBFNetwork) algorithm. The findings showed that the model could predict GDM. This allows doctors to step in and reduce unfavourable outcomes for mothers and babies [65].

## 8. Challenges and Future Directions

Enhancing patient outcomes, predicting outbreaks, and providing deep insights not only help with any kind of prevention but also open doors to an increased quality of life. Ultimately, big data shall transform healthcare to deliver improved health to individuals and communities as well [66].

One of the challenges it brings about is that dealing with data in many cases proves easier said than done, from the simple task of catching/collecting and storing to more complex tasks such as analysis and visualization. There are many issues regarding data structure—big data, instead of being informed and open, is partly dispersed, scattered, and rarely consistent from source to origin, with data security being a major issue in storage and transfers (especially the costs associated with securing, storing, and transferring unstructured data), and there is a shortage of managerial and analytical skills, particularly for real-time analytics [67].

The extensive collection of medical data from a variety of sources poses a significant challenge for data scientists regarding its integration and practical application. To advance healthcare, it is crucial to merge bioinformatics, health informatics, and analytics to foster personalized and effective treatments. The arrival of big data has already resulted in remarkable progress in healthcare, spanning from data management to drug discovery for intricate diseases. Instead of replacing specialists, big data will bolster advancements in healthcare, redirecting attention towards personalized care and predictive analytics. Future applications include forecasting health outcomes, predicting epidemics, and discovering new biomarkers and therapies, ultimately improving quality of life.

The ethical considerations associated with big data in healthcare are highly complex and multidimensional, covering a range of issues that significantly affect individuals, institutions, and the overall society.

**1.** **Informed Consent**: Individuals may provide consent for the collection and use of their data without fully understanding the potential future applications, particularly as data can be repurposed, aggregated, and shared across diverse platforms [68] Google’s Project Nightingale collected healthcare data from millions without patient consent, leading to public backlash and calls for stricter transparency and consent protocols. IBM’s AI ethics initiatives emphasize transparency and explainability, requiring that AI decision-making processes be understandable to stakeholders (7 Essential Data Ethics Examples for Businesses in 2025).**2.** **Privacy and Confidentiality**: Protecting the privacy and confidentiality of sensitive health data is a primary concern in big data applications [69]. While anonymization and de-identification techniques are routinely used, these methods are not foolproof, as advances in data linkage and re-identification techniques have demonstrated that individuals can still be identified by combining disparate data sources [70].**3.** **Data Ownership and Control:** The issue of data ownership is a central ethical challenge in the big data landscape [71]. This raises questions regarding the rights of data subjects and whether they should be entitled to a share of the benefits that result from the use of their data, particularly in cases where institutions or corporations derive financial or intellectual gains.**4.** **Equity and the Big Data Divide:** The capacity to harness big data for healthcare innovation is disproportionately concentrated among institutions with advanced technological infrastructure, deep financial resources, and sophisticated analytical expertise. This concentration creates a “big data divide”, wherein institutions and populations with fewer resources may be left behind, exacerbating existing health disparities [72].**5.** **Epistemological Challenges:** The vast scale of big data in healthcare creates an over-reliance on correlation-driven insights, often without a clear understanding of the underlying causal mechanisms. This presents significant epistemological challenges, as decisions based on superficial correlations may lead to erroneous conclusions and suboptimal interventions [73].

Table 4 presents the key challenges of big data analytics (BDA) in healthcare, along with relevant examples.

## 9. Future Directions: Balancing Core Considerations

A consistent patient consent framework is essential. Patient consent drives data availability, with evidence suggesting that individuals are generally willing to share their health data when approached [80]. The flexibility of HIPAA rules allows consent forms to be combined across research studies, making the process more efficient [81]. Dynamic consent forms tailored to the specific needs of each study and patient record could streamline this process, only including relevant elements [82].

Data de-identification is yet another crucial security measure. Despite worries concerning the ease of re-identification, research suggests that attacks on fully HIPAA-compliant de-identified datasets are relatively uncommon [83]. The big data platforms make de-identified data more useful for research by facilitating the statistical analysis of widespread datasets and longitudinal studies without disclosing private health information [83]. Healthcare institutions should create reliable rules for data usage that shield people from potential damage in situations where data segmentation is inadequate. Encryption and authorization verification processes further enhance data security while encouraging transparency and trust in big data research.

## 10. Conclusions

The dawn of a new era of BDA has enormous potential for providing physicians and other healthcare professionals with a personalized and individualized approach to patient treatment based on vast amounts of comprehensive data. However, it is apparent that BDA will advance as a new style of practising medicine.

Biomedical and healthcare tools like genomics, biometric sensors, and smartphone apps now generate vast amounts of data, necessitating a better understanding of how to leverage this information. Integrating big data from electronic health records (EHRs), electronic medical records (EMRs), and other sources helps refine prognostic models and treatment strategies. Healthcare analytics firms aim to lower costs, enhance clinical decision support (CDS) systems, and develop effective platforms while facing challenges related to data privacy and security. The vast data pool from healthcare and biomedical research has led to improved disease diagnosis, treatment, and prevention. Advances in computing, including supercomputers and quantum computing, accelerate the extraction of actionable insights from big data. Despite infrastructure challenges, BDA continue to drive innovation in clinical practices and personalized healthcare.

## Figures and Tables

**Figure 1 bioengineering-12-00463-f001:**
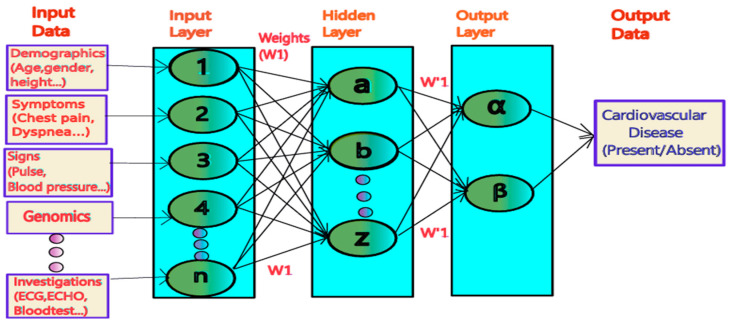
An important simplistic application of ML.

**Table 1 bioengineering-12-00463-t001:** Various aspects of big data analytics.

Descriptive Analytics	Aims to Examine Past Datasets for Patterns and Trends
Predictive analytics	Aims to predict likely outcomes and make evidence-based forecasts using historical data
Prescriptive analytics	Utilizes data from diverse sources, such as statistical analyses, machine learning algorithms, and data mining techniques, to predict potential future outcomes and determine the most optimal course of action
Diagnostic analytics	Analysing historical and real-time data to identify the underlying causes

**Table 2 bioengineering-12-00463-t002:** Comparison of the various models using big data and other resources. “✓” stands for available feature while “×” stands for feature not available.

Related Works	Big Data	Map Reduce	Cloud	Cardiac Healthcare Data	ECG Data
Sahoo P.K. et al., 2018 [14]	✓	×	✓	✓	×
Sahoo P.K. et al., 2016 [15]	✓	✓	✓	✓	×
Manimurugun et al., 2022 [16]	✓	×	✓	✓	×
Choi et al., 2020 [17]	✓	✓	×	×	×
Safa et al., 2023 [18]	✓	×	×	✓	×
Mohapatra et al., 2024 [19]	✓	✓	✓	✓	✓

**Table 3 bioengineering-12-00463-t003:** Some of the models used for cardiac data analysis with Accuracy rates.

Authors	Dataset	Algorithm Type	Analysis	Number of Features	Accuracy (%)
Srinivasan et al., 2023 [13]	UCI respository	Learning vector quantization (LVQ)	Classification	10	98
AI Bataineh & Manacek 2022 [33]	Heart disease	Multilayer Perceptron (MLP) + PSO	Classification	13	84
AI Bataineh & Manacek 2022 [33]	Heart disease	Recurrent neural network (RNN) + long short-term memory (LSTM)	Classification	14	95
Pathan M.S et al., 2022 [34] (77)	Cardiovascular Disease (CVD) and Framingham	MLP, support vector classifier	Classification	12 (CVD)	74 (CVD)
				11 (Fram)	71 (Fram)
Ozcan M et al., 2023 [35]	Cleveland, Hungarian, Switzerland, Long Beach VA Stalog Dataset	Classification and regression tree (CART)	Classification and Regression	11	87
Verma L et. al., 2016 [36]	Department of Cardiology, IGMC	Multinominal logistic regression (MLR)	Classification	26	98

**Table 4 bioengineering-12-00463-t004:** Challenges of BDA with examples.

Challenge with BDA Usage	Descriptions	Scientific Evidence/Implications	Citation
**Data Privacy and Security**	Data safety, patient identifiers, and data breaches might raise concerns about compliance with regulatory agencies.	Studies show that unauthorized access to health data can lead to loss of trust, legal consequences, and delays in adopting analytics. Privacy-preserving models (e.g., federated learning) are being explored.	[74]
**Integration of Data Sources**	BDA uses heterogeneous data sources that combine information on demographics, clinical, anthropometric, lifestyle and risk factors, genomics, metabolomics, and imaging tools. This complexity adds variations in format.	Research highlights difficulties in achieving interoperability across electronic health records (EHRs), devices, and databases. Standards like FHIR are being developed to address this.	[75]
**Infrastructure Costs**	BDA can be a cost-sensitive technique due to the requirements of high computational power, storage, and skilled professionals.	Studies estimate significant upfront and ongoing costs for hospitals and research institutions. Cloud-based solutions can help mitigate infrastructure burdens but may raise additional concerns about data governance.	[76]
**Algorithm Bias and Accuracy**	Data gaps, inaccurate datasets, and poorly coded heterogeneous data can generate misleading and biased algorithms. Often, CVD’s determinants are contextual and a lack of Indigenous data might build inaccurate models for various populations.	Evidence shows that the underrepresentation of certain populations in datasets can lead to biased outcomes. Initiatives to increase diversity in data collection and ethical AI practices are essential.	[77]
**Ethical and Legal Challenges**	Ambiguities around ownership, consent, and the ethical use of patient data complicate the deployment of analytics in healthcare.	Researchers highlight the importance of clear legal frameworks and ethical guidelines. For example, consent models for the secondary use of data in research remain a contested issue.	[78]
**Resource requirements**	BDA is an emerging area requiring expertise from diverse fields (for BDA in the area of CVD: data scientists, clinicians).	Reports emphasize the shortage of professionals trained in both healthcare and data analytics. Education and training programs integrating both domains are critical for capacity building.	[79]

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
