# Peer review of "Revolutionizing Utility of Big Data Analytics in Personalized Cardiovascular Healthcare"

_bioengineering, 2025, doi:10.3390/bioengineering12050463_

Round 1

Reviewer 1 Report

Comments and Suggestions for Authors
  1. The author affiliations should be written in full, including the complete institution name, city, postcode, and country, without job titles such as "student" or "associate professor." These titles should be placed in the "Corresponding Author" section, if applicable. For example, the affiliation should read something like: "Department of Cardiology, U.N. Mehta Institute of Cardiology and Research Centre, BJ Medical College, Ahmedabad, Gujarat, 380013, India. Email: kamalsharma1975@gmail.com." The corresponding author should be listed separately after the affiliations, with their contact details provided.
  2. The title, "Revolutionizing Cardiovascular Healthcare Using Big Data Analytics - Directions and Challenges in Personalized Medicine," does not fully reflect the real contribution of the study and should be more specific about the focus of Big Data in personalized cardiovascular healthcare.
  3. The abstract serves as a nice introduction to BDA in CVD but does not clearly reflect the original contributions of your work. It describes BDA uses in health care and CVD management scenarios but without details of the innovative methods, frameworks or results developed by the authors. To do this, the abstract should better emphasize the concrete contributions of your research, such as original methods you may have had for integrating different data types (e.g. genomic data, proteomic data, data on lifestyle characteristics or outcomes) or novel models of prediction that you may have developed. It should also provide a short recap of the key results and how they will be advancing the field, providing potential translational information about how your work might make a difference in the management of the patient or how the disease is understood. Such revisions will keep your abstract focused on the central innovations of your study and how they contribute to the field.
  4. The paper gives a systematic and broad perspective of BDA. It can be complemented with more specific illustrations to elaborate how BDA can be implemented in real-life scenarios with cardiovascular diseases. This article explains how BDA facilitates personalized medicine in cardiovascular diseases; however, more explanations with examples in integrating data from heterogeneous information sources (e.g., EHRs, genomics, and wearable technologies) could be added.
  5. Less documentation should be included about how various data types are combined for predictive modeling (data preprocessing and integration methods). The paper could include more explanation of how the multimodal data fusion is achieved, particularly in terms of resolving conflicts between data sources.
  6. Insufficient discussion of machine learning (ML) and artificial intelligence (AI) methods is found in predictive modeling literature, which fails to address risks of bias in CVD datasets. This could lead to skewed predictions with the underrepresentation of demographic groups, which needs to be addressed.
  7. This paper touches on the issues of machine learning, especially the interpretability and transparency of machine learning models. In medical machine learning, more research is needed on preventing overfitting and interpreting complex AI models.
  8. This paper is not a thorough review of recent literature. It can be a lot more relevant and current if up-to-date studies are included, especially the new advances in BDA and AI in cardiovascular health. Identifying recent work with a forward-thinking mindset is the way to align the paper with the trend of the research.
  9. I have carefully chosen the cited references based on the specific gaps identified in the manuscript, particularly in the introduction and literature review sections. These references provide essential background, address the missing foundational context, and enhance the discussion with relevant prior work. Additionally, I confirm that these are not self-citations but rather the most suitable and high-impact sources that align with the study’s objectives. 
  • Zhang, Y., Zhu, L., Li, X., Ge, C., Pei, W., Zhang, M.,... Lv, K. (2024). M2 macrophage exosome-derived lncRNA AK083884 protects mice from CVB3-induced viral myocarditis through regulating PKM2/HIF-1α axis mediated metabolic reprogramming of macrophages. Redox Biology, 69, 103016. doi: https://doi.org/10.1016/j.redox.2023.103016
  • Li, L., Li, J., Zhong, M., Wu, Z., Wan, S., Li, X.,... Lv, K. (2025). Nanozyme-enhanced tyramine signal amplification probe for preamplification-free myocarditis-related miRNAs detection. Chemical Engineering Journal, 503, 158093. doi: https://doi.org/10.1016/j.cej.2024.158093

Author Response

Comments 1 -  The author affiliations should be written in full, including the complete institution name, city, postcode, and country, without job titles such as "student" or "associate professor." These titles should be placed in the "Corresponding Author" section, if applicable. For example, the affiliation should read something like: "Department of Cardiology, U.N. Mehta Institute of Cardiology and Research Centre, BJ Medical College, Ahmedabad, Gujarat, 380013, India. Email: kamalsharma1975@gmail.com." The corresponding author should be listed separately after the affiliations, with their contact details provided.

Response 1- We thank you for bringing this to our notice and now we have corrected the author affiliations as suggested. Thanks a lot.

Comment 2- The title, "Revolutionizing Cardiovascular Healthcare Using Big Data Analytics - Directions and Challenges in Personalized Medicine," does not fully reflect the real contribution of the study and should be more specific about the focus of Big Data in personalized cardiovascular healthcare.

Response 2- Thanks for pointing out and now we have revised the title to "Revolutionizing utility of Big Data Analytics in Personalized Cardiovascular healthcare "

Comment 3- The abstract serves as a nice introduction to BDA in CVD but does not clearly reflect the original contributions of your work. It describes BDA uses in health care and CVD management scenarios but without details of the innovative methods, frameworks or results developed by the authors. To do this, the abstract should better emphasize the concrete contributions of your research, such as original methods you may have had for integrating different data types (e.g. genomic data, proteomic data, data on lifestyle characteristics or outcomes) or novel models of prediction that you may have developed. It should also provide a short recap of the key results and how they will be advancing the field, providing potential translational information about how your work might make a difference in the management of the patient or how the disease is understood. Such revisions will keep your abstract focused on the central innovations of your study and how they contribute to the field.

Response 3 - Thanks to bring this important point to mention the methods and our contribution though this review article that needs to be summarized in the abstract. In view of these important aspect to be incorporated in the abstract now we have mentioned "We searched PubMed and published BDA research using Google and Cochrane search engines to evaluate existing models of BDA that have been used for CVD prediction models. We critically analyze pitfalls and advantages of various BDA models using Artificial Intelligence (AI), Machine learning (ML) and Artificial neural networks (ANN). "

Comments 4- The paper gives a systematic and broad perspective of BDA. It can be complemented with more specific illustrations to elaborate how BDA can be implemented in real-life scenarios with cardiovascular diseases. This article explains how BDA facilitates personalized medicine in cardiovascular diseases; however, more explanations with examples in integrating data from heterogeneous information sources (e.g., EHRs, genomics, and wearable technologies) could be added.

Response 4- This is a very important and insightful comment from the esteemed reviewer and we thank you for your input. Though we have mentioned many other examples but including examples of EHR and wearable technologies was missing which we have now included at Line 153-156 and line 203-206 as following- "The QRISK4 and PREVENT models leverage EHR data combined with social determinants of health to predict 10- to 30-year cardiovascular risks. These models exclude race as a predictor and instead use social deprivation indices (income, education, housing) to address disparities. (Ming-Lung Tsai et.al 2024) " and " The Apple Heart Study analyzed data from 400,000 participants using smartwatches to detect irregular pulses. This wearable-derived data when integrated with EHRs for validating of atrial fibrillation detection, achieved 84% concordance with clinical ECG findings. (Lovedeep Singh Dhingra et.al 2015)". Thanks again

Comments 5- Less documentation should be included about how various data types are combined for predictive modeling (data preprocessing and integration methods). The paper could include more explanation of how the multimodal data fusion is achieved, particularly in terms of resolving conflicts between data sources.

Response 5- Thanks for insightful comments. we have now trimmed the predictive modelling methods as in Line 143-146. Multimodal data fusion examples are already included as mentioned in comment 4 with line 153-156 and line 203-206 as mentioned above.

Comment 6- Insufficient discussion of machine learning (ML) and artificial intelligence (AI) methods is found in predictive modelling literature, which fails to address risks of bias in CVD datasets. This could lead to skewed predictions with the underrepresentation of demographic groups, which needs to be addressed.

Response 6- Thanks for pointing out the need for further discussion on addressing need for data level and algorithmic approaches to refine and avoid development of biases for underrepresentation of demographic groups. We have now added the same from line 178 to 186 as "Data-Level Approaches to address the demographic biases and underrepresentation can be addressed are Resampling by adjusting training data balance by oversampling minority groups or under sampling majority groups. Feature Engineering which uses. Hybrid models like ANN-GA (86.4% accuracy) use genetic algorithms to select relevant features, reducing redundant attributes that may introduce bias (Buhari Ugbede Umar et.al 2024). Algorithmic Adjustments include Fairness-Aware Modelling Techniques like adversarial debiasing and post-processing (e.g., equalized odds) and reduced false favourable rates across genders. Interpretability Tools: SHAP and LIME explain model decisions, helping identify biased feature contributions. (Md Abu Safin et.al. 2024)"

Comment 7-This paper touches on the issues of machine learning, especially the interpretability and transparency of machine learning models. In medical machine learning, more research is needed on preventing overfitting and interpreting complex AI models.

Response 7 - Thanks for the insightful comment. Now the revised format addresses not only AI but also ML and ANN models with many new references and citations as mentioned in the draft. 

Comment 8-This paper is not a thorough review of recent literature. It can be a lot more relevant and current if up-to-date studies are included, especially the new advances in BDA and AI in cardiovascular health. Identifying recent work with a forward-thinking mindset is the way to align the paper with the trend of the research.

Response 8- Thanks for updating the same. The new literature and references are now updated right up to 2025 and includes new and recent updates not only from Cardiology journals (JAHA) but also from AI and biomedicine as highlighted in the manuscript and references viz. line 679-696 and is as follows - 

  1. Ming‐Lung Tsai, MD, Kuan‐Fu Chen, MD, (2025) Harnessing Electronic Health Records and Artificial Intelligence for Enhanced Cardiovascular Risk Prediction: A Comprehensive Review Journal of the American Heart Association, Volume 14, Number 6 , https://doi.org/10.1161/JAHA.124.036946
  2. Dhingra LS, Shen M, Mangla A, Khera R. Cardiovascular Care Innovation through Data-Driven Discoveries in the Electronic Health Record. Am J Cardiol. 2023 Sep 15;203:136-148. doi: 10.1016/j.amjcard.2023.06.104. Epub 2023 Jul 25. PMID: 37499593; PMCID: PMC10865722.
  3. Buhari Ugbede Umar, Lukman Adewale Ajao, Eustace Mananyi Dogo, Falilat Jumoke Ajao, Micheal Atama. Artificial intelligence model for prediction of cardiovascular disease: An empirical study. Artificial Intelligence in Health2024, 1(1), 42–56. https://doi.org/10.36922/aih.1746
  4. Sufian MA, Alsadder L, Hamzi W, Zaman S, Sagar ASMS, Hamzi B. Mitigating Algorithmic Bias in AI-Driven Cardiovascular Imaging for Fairer Diagnostics. Diagnostics (Basel). 2024 Nov 27;14(23):2675. doi: 10.3390/diagnostics14232675. PMID: 39682584; PMCID: PMC11640708.
  5. Zhang, Y., Zhu, L., Li, X., Ge, C., Pei, W., Zhang, M.,... Lv, K. (2024). M2 macrophage exosome-derived lncRNA AK083884 protects mice from CVB3-induced viral myocarditis through regulating PKM2/HIF-1α axis mediated metabolic reprogramming of macrophages. Redox Biology, 69, 103016. doi: https://doi.org/10.1016/j.redox.2023.103016
  6. Li, L., Li, J., Zhong, M., Wu, Z., Wan, S., Li, X.,... Lv, K. (2025). Nanozyme-enhanced tyramine signal amplification probe for preamplification-free myocarditis-related miRNAs detection. Chemical Engineering Journal, 503, 158093. doi: https://doi.org/10.1016/j.cej.2024.158093

Comment 9- I have carefully chosen the cited references based on the specific gaps identified in the manuscript, particularly in the introduction and literature review sections. These references provide essential background, address the missing foundational context, and enhance the discussion with relevant prior work. Additionally, I confirm that these are not self-citations but rather the most suitable and high-impact sources that align with the study’s objectives. 

Response 9- Thanks for pointing 2 articles of molecular biology. we have now mentioned them in the review of literature as well as in the references as in Line 667-684. The manuscript now mentions "Development of lncRNA AK083884 and Nanozyme-enhanced tyramine signal amplification probe have shown protection from viral myocarditis and it’s genetic implications can have futuristic value in preventing the same.(Zhang Y et.al 2024, Li. L 2025) "

Thanks for your valuable time evaluating the draft and we hope that we have addressed all your concerns to the fullest of your satisfaction. Thanks again.

Reviewer 2 Report

Comments and Suggestions for Authors

This review is devoted to new methods in the field of cardiovascular medicine using big data analytics in personalized healthcare.

There are some comments on the manuscript.

Abstract. In my opinion, it is necessary to reduce general words about big data and add specifics about healthcare - the range of tasks and goals.

Introduction. Although this manuscript is a review, it is still necessary to add about the objectives of the review and the last paragraph about the structure of the article.

Section 2. In the title (line 58) there is an extra character - :

Figure 1. Here it clearly does not reflect the state of machine learning well. The authors should replace the picture with a more meaningful one. It is impossible to squeeze all the methods of machine learning into a three-layer neural network.

In my opinion, it is possible to add a section devoted to modern data collection technologies - ECG sensors, watches and smartphones.

The Discussion section would be necessary, since it is necessary to combine general conclusions and discussions.

Overall, the review is interesting, it can be recommended after revision.

Author Response

Comments 1- This review is devoted to new methods in the field of cardiovascular medicine using big data analytics in personalized healthcare.

Response 1- Thanks for your appreciation and kind words.

Comments 2-There are some comments on the manuscript.

Abstract. In my opinion, it is necessary to reduce general words about big data and add specifics about healthcare - the range of tasks and goals.

Response 2- We have reduced work about big data and elaborated it on healthcare front. We have deleted lines “BDA has potential applications in healthcare to transform patient care and manage diseases by allowing personalized treatment plans and enhancing clinical outcomes” and added few more lines for about healthcare.

Comment 3- Introduction. Although this manuscript is a review, it is still necessary to add about the objectives of the review and the last paragraph about the structure of the article.

Section 2. In the title (line 58) there is an extra character - :

Response 3- Thanks for bringing this to our notice. We have added objective now in the manuscript as “The objective of this article is to evaluate and compare the available BDA algorithms along with AI, ML and PL in the current era with their pitfalls and how these can be further integrated to refine CVD diagnostics and management.” Line 57-59. The extra character is now deleted in the section 2.

Comment 4 - Figure 1. Here it clearly does not reflect the state of machine learning well. The authors should replace the picture with a more meaningful one. It is impossible to squeeze all the methods of machine learning into a three-layer neural network.

Response 4 – Thanks for the insightful comments. We have now modified the figure to nth variables and changed title as well the description of the figure and mentioned “Too simplistically  put, based on the scientifically and statistically significant input datasets apart from the outliers; after being incorporated in the input layer, a weighted analysis identifies the contribution of each variable in the hidden layer which further on weighted analysis in output layer gives output data as presence or absence of cardiovascular disease as shown in Figure 1.” In lines 144 -149. This figure is based on Dykstra et al., 2022 and we have kept it simplified so that those without Data sciences background can comprehend the method involved. Thanks

Comment 5 - In my opinion, it is possible to add a section devoted to modern data collection technologies - ECG sensors, watches and smartphones. The Discussion section would be necessary, since it is necessary to combine general conclusions and discussions

Response 5- Thanks for comments. Paragraphs are added about wearable technologies as suggested without losing focus on big data which is the objective of the paper. “The Apple Heart Study analyzed data from 400,000 participants using smartwatches to detect irregular pulses. This wearable-derived data when integrated with EHRs for validating of atrial fibrillation detection, achieved 84% concordance with clinical ECG findings. (Lovedeep Singh Dhingra et.al)” Line 203-206

Several preprocessing steps such as normalisation and feature selection have demonstrated how this can help cardiovascular disease care process more accessible and precise by generating personalised recommendations based on patient's data acquired from wearable low-cost sensors or other devices but also the disruptive changes due to big-data analytics and machine learning on advancing patient disease management and prescription plans (Lalani et al., 2021)-Line 342-347

Comment 6- Overall, the review is interesting, it can be recommended after revision.

Response 6- Thanks for your appreciation of out work.

Reviewer 3 Report

Comments and Suggestions for Authors

The topic is very important and timely — it talks about using big data analytics in cardiovascular medicine and how it can help make personalized treatment possible. This is a relevant topic because heart diseases are very common, and we need better and faster ways to diagnose and treat them. The use of machine learning, wearable devices, genetic information, and real-time monitoring is a very interesting and future-focused idea.

The authors have collected a lot of information from many research studies and included different examples and tables to support their points. I especially liked the part where they discussed how real-time data from smartwatches or health apps can help doctors give faster and better treatment. It shows the value of technology in healthcare clearly.

But I think the paper also has some major problems. First of all, it is very long and contains too much technical detail, which makes it difficult to follow sometimes — especially for readers who are not experts in this area. Some sections feel repetitive, for example, the definition and explanation of big data comes again and again in different places. The same phrases like "big data analytics" are used too often. The paper would benefit from being shorter and more to the point.

Second, the writing style can be improved. Some sentences are too long and have complicated words, and grammar mistakes are also present. The ideas should be explained more simply, so that non-specialist readers can also understand. It would help if technical terms are defined clearly and not overused.

Third, I found the structure of the paper not very well organized. Sometimes it is difficult to understand where the background ends and where the discussion or results begin. Also, the tables are placed without enough explanation around them. There are too many references listed together in some places, which feels more like a bibliography dump rather than thoughtful discussion.

I also feel that some claims in the paper are too general. For example, saying that AI or BDA can improve treatment or reduce disease burden is not enough — the authors should give more specific data, numbers, or real-life examples to support those points. The ethical issues are discussed, but only in theory. I think adding real-world ethical challenges and how they were handled would make the paper stronger.

Another thing that can improve the paper is including more visual elements. Right now, it’s mostly text and tables. Some figures or charts could help explain complex models and methods more clearly.

In summary, I think this paper covers an important and valuable topic with many strong ideas and a lot of effort. But it also needs major revision. The content should be made shorter, the language simplified, structure improved, and claims supported by stronger examples. If these changes are made, the paper can be much more effective and useful to the readers.

Author Response

Comments 1-

The topic is very important and timely — it talks about using big data analytics in cardiovascular medicine and how it can help make personalized treatment possible. This is a relevant topic because heart diseases are very common, and we need better and faster ways to diagnose and treat them. The use of machine learning, wearable devices, genetic information, and real-time monitoring is a very interesting and future-focused idea.

The authors have collected a lot of information from many research studies and included different examples and tables to support their points. I especially liked the part where they discussed how real-time data from smartwatches or health apps can help doctors give faster and better treatment. It shows the value of technology in healthcare clearly.

Response 1 – Thanks a lot for your appreciation of out work. Regards

Comments 2- But I think the paper also has some major problems. First of all, it is very long and contains too much technical detail, which makes it difficult to follow sometimes — especially for readers who are not experts in this area. Some sections feel repetitive, for example, the definition and explanation of big data comes again and again in different places. The same phrases like "big data analytics" are used too often. The paper would benefit from being shorter and more to the point.

Response 2- Thanks for your comments. We have made changes in the draft and deleted many lines to shorten the draft and added few recent updates to make it interesting. We have now replaced Big data analytics with BDA and now the term is used only with tables and titles in the manuscript.

Comments 3 - Second, the writing style can be improved. Some sentences are too long and have complicated words, and grammar mistakes are also present. The ideas should be explained more simply, so that non-specialist readers can also understand. It would help if technical terms are defined clearly and not overused.

Response 3- Thanks for important input. We have now revised the manuscript from the linguistic as well as grammatical perspective with native English-Speaking language expert and simplified the draft all across the manuscript.

Comments 4 –

Third, I found the structure of the paper not very well organized. Sometimes it is difficult to understand where the background ends and where the discussion or results begin. Also, the tables are placed without enough explanation around them. There are too many references listed together in some places, which feels more like a bibliography dump rather than thoughtful discussion.

Response 4- Thanks for your insightful comments. We have reformatted the draft now to maintain the flow. We agree that there are many references but being a review article, we wanted to present all the latest research pertaining to BDA in CVD. Keeping in line of the constructive suggestions we have now deleted a few of the references.

Comments 5 –

I also feel that some claims in the paper are too general. For example, saying that AI or BDA can improve treatment or reduce disease burden is not enough — the authors should give more specific data, numbers, or real-life examples to support those points. The ethical issues are discussed, but only in theory. I think adding real-world ethical challenges and how they were handled would make the paper stronger.

Response 5- Thanks for pointing this important aspect. In view of your suggestions, we have now added examples as follows.

The QRISK4 and PREVENT models leverage EHR data combined with social determinants of health to predict 10- to 30-year cardiovascular risks. These models exclude race as a predictor and instead use social deprivation indices (income, education, housing) to address disparities. (Ming-Lung Tsai et.al 2024)- LINE 153-156.

The Apple Heart Study analyzed data from 400,000 participants using smartwatches to detect irregular pulses. This wearable-derived data when integrated with EHRs for validating of atrial fibrillation detection, achieved 84% concordance with clinical ECG findings. (Lovedeep Singh Dhingra et.al)  Line-203-206

The ethical considerations exist for use of BDA and we have covered the same.The same is mentioned as “ Google’s Project Nightingale collected healthcare data from millions without patient consent, leading to public backlash and calls for stricter transparency and consent protocols. IBM’s AI ethics initiatives emphasize transparency and explainability, requiring that AI decision-making processes be understandable to stakeholders. (7 Essential Data Ethics Examples for Businesses in 2025)” LINE-428-431

Comments 6-

Another thing that can improve the paper is including more visual elements. Right now, it’s mostly text and tables. Some figures or charts could help explain complex models and methods more clearly.

Response 6-

Thanks for your comments. We have modified the tables and figure for more clarity

Comments 7-

In summary, I think this paper covers an important and valuable topic with many strong ideas and a lot of effort. But it also needs major revision. The content should be made shorter, the language simplified, structure improved, and claims supported by stronger examples. If these changes are made, the paper can be much more effective and useful to the readers.

Response 7- Thanks for your comments and encouragement. We have made changes as per your suggestions and hope that the revised version now is suitable as the scientific discourse worth consideration. Thanks again

Round 2

Reviewer 1 Report

Comments and Suggestions for Authors

Accept

Reviewer 3 Report

Comments and Suggestions for Authors

I accept the paper in its present form.